# Bimodality in Ensemble Forecasts of 2-Meter Temperature: Identification

Cameron Bertossa[1], Peter Hitchcock[1], Arthur DeGaetano[1], and Riwal Plougonven[2]

[1]Dept. Earth and Atmospheric Sciences, Cornell University, Ithaca, New York, USA
[2]LMD-IPSL, Ecole Polytechnique, Institut Polytechnique de Paris, ENS, PSL Research University, Sorbonne Université, CNRS, Paris, France

**Correspondence:** Cameron Bertossa (cdb227@cornell.edu)

**Abstract.** Bimodality and other types of non-Gaussianity arise in ensemble forecasts of the atmosphere as a result of non-linear spread across ensemble members. In this paper, bimodality in 50-member ECMWF ENS-extended ensemble forecasts is identified and characterized. Forecasts of 2-meter temperature are found to exhibit widespread bimodality well over a derived false-positive rate. In some regions bimodality occurs in excess of 30% of forecasts, with the largest rates occurring during lead times of 2 to 3 weeks. Bimodality occurs more frequently in the winter hemisphere with indications of baroclinicity being a factor to its development. Additionally, bimodality is more common over the ocean, especially the polar oceans, which may indicate development caused by boundary conditions (such as sea ice). Near the equatorial region, bimodality remains common during either season and follows similar patterns to the intertropical convergence zone (ITCZ) suggesting convection as a possible source for its development. Over some continental regions the modes of the forecasts are separated by up to 15 $^{o}$C. The probability density for the modes can be up to four times greater than at the minimum between the modes, which lies near the ensemble mean. The widespread presence of such bimodality has potentially important implications for decision makers acting on these forecasts. Bimodality also has implications for assessing forecast skill and for statistical post-processing: several commonly used skill scoring methods and ensemble dressing methods are found to perform poorly in the presence of bimodality, suggesting the need for improvements in how non-Gaussian ensemble forecasts are evaluated.

## 1 Introduction

The atmosphere is a highly chaotic system that is not easily predicted. An important development has been the use of ensemble forecasts in order to develop a probabilistic viewpoint of the future state of the atmosphere. An ensemble is a grouping of multiple forecasts initialized at the same time, each slightly perturbed relative to one another. These perturbations result in the divergence over time of the individual members (Lorenz, 1963), producing a diversity of 'point' forecasts from which a probabilistic forecast can be generated. A great deal of attention has focused on the use of the ensemble mean and variance of the ensemble, explicitly or implicitly assuming that the distributions are Gaussian. While there are technical and practical reasons for doing so, treating all ensemble forecasts as such for the entirety of their predictions is not always appropriate and may miss valuable distributional information available, especially in larger ensembles. This work focuses explicitly on one specific type of non-Gaussianity: bimodality.

Bimodal distributions occur when some of a forecast's ensemble members form a distinct cluster, separate from the rest of the ensemble. Depending on the methodology, this can result in generating a corresponding probability density function (PDF) that contains two (or more) modes. This paper quantifies the presence of bimodality in ensemble forecasts of 2-meter temperature from the extended 46-day, 50-member forecasts produced by the European Centre for Medium-range Weather Forecasts (ECMWF). The focus of this work is on assessing the frequency and character of bimodality in these forecasts.

Forecast errors arise both from our imperfect knowledge of the initial conditions, and from imperfect forecast models. The growth of these errors is not uniform (Palmer, 2000; Leutbecher and Palmer, 2008), so that there is great practical value in assessing the certainty associated with any given forecast. Although other approaches exist, ensembles have become the method of choice to produce these probabilistic forecasts (Buizza, 2018). By perturbing each member in some appropriate way, the range of outcomes across the ensemble can be used to construct a probablistic assessment of plausible future outcomes.

However, there are many challenges in constructing these probabilistic forecasts from a raw ensemble. The phase space of possible future atmospheric states is extremely high-dimensional, so that a full representation of the 'true' multivariate PDF representing a given forecast is unattainable for ensembles of finite size (see Kalnay, 2019, and references therein).

The perturbations applied to each member must be constructed appropriately, so that the spread across the ensemble can nonetheless capture the most salient parts of this 'true' distribution. In practice, a key measure is the spread of the ensemble.

Historically, 'underdispersion' is a major difficulty, meaning that validating observations are often found outside the range of the forecast ensemble (Wang et al., 2018). Considerable effort has gone into appropriate initialization schemes to mitigate this issue; for instance, the ECMWF forecasts use singular vectors over a defined time interval, chosen to maximize ensemble spread (Leutbecher and Palmer, 2008).

Several examples of ECMWF ensemble forecasts of 2-meter temperature at selected grid points are shown in Fig. 1. The

gray lines in Fig. 1(a) show the 50 individual ensemble members of the same forecast in Fig. 1(b). The ensemble members are initially tightly clustered around the mean when the differences caused by the initialization perturbations are small; the members then disperse over time. The validation observations for each forecast are represented with yellow lines. Observations are based on ERA-Interim reanalysis (Dee et al., 2011). The tendency of forecasts to be underdispersed can be seen in Fig. 1(a,b); this feature is prevalent in the first 10 days of the forecast. The non-uniform nature of forecast spread (Palmer,

2000; Leutbecher and Palmer, 2008) is also apparent: in one case (Fig. 1(c)) the spread remains relatively small throughout the forecast, whereas in another forecast for the same location (Fig. 1(d)) the spread grows much more rapidly. The spread is non-uniform in space as well; compare Fig. 1(b) and Fig. 1(d), which show two forecasts initialized on the same forecast date but at different locations.

Figure 1(d) presents a case in which the ensemble spread is strongly non-Gaussian. The shading indicates a forecast PDF

generated by a kernel density estimate (KDE) (Wilks, 2011), suggesting the presence of bimodality; the lead times at which this bimodality is present according to a statistical test (described in Sect. 2) are indicated with green dashes. In these cases the ensemble mean lies near a local minimum in the bimodal forecast PDF. The bimodality of the forecast suggests instead that there are two potential scenarios that may occur, and that the spread about these two modes is considerably less than the standard deviation of the ensemble as a whole might suggest. Those lead times whose validation observations would have been

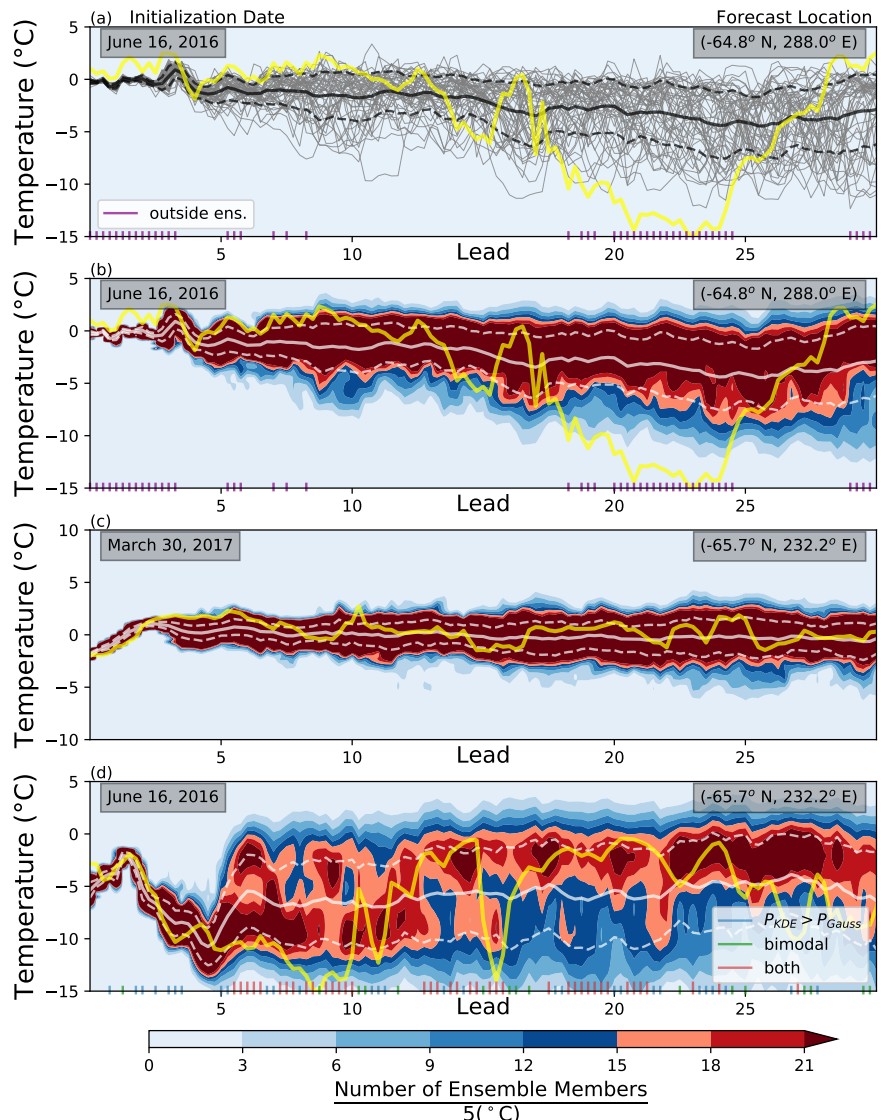

**Figure 1.** The time evolution in days of three different ECMWF ensembles and their respective observation values. Forecasts are 6-hourly 2-meter temperature values. Forecast initialization dates are in the upper left, and forecast location is in the upper right. (a) The individual ensemble members as a function of lead time in gray. (b-d) The kernel density estimate (KDE) of the ensemble distribution as colored contours, where darker shades of red indicate a high number of ensemble members near that temperature ($\pm$ 2.5 $^o$C). Note that (b) depicts the same forecast as (a). The mean and one standard deviation of the distribution at a given lead time are in white. The yellow line represent the validation at a given lead time. Purple axis ticks in (a,b) represent observation values that lie outside the warmest or coolest ensemble member at that lead time. Blue tick marks in (d) represent a KDE that predicts the validation with a greater probability than a Gaussian fit, KDE distributions who are bimodal are marked with green, and those lead times when these two conditions coincide are marked in red.

predicted with a higher probability with a KDE PDF compared to a Gaussian PDF occurs in quite a few of the lead times (blue dashes). These lead times also commonly align with bimodality in the distribution (red dashes). This work shows that such bimodality in forecasts of 2-meter temperature are in fact reasonably common.

## 1.1 Why Identify Bimodality in Forecasts?

There are several motivations for considering whether episodes of bimodality similar to that suggested by Fig. 1(d) occur

regularly. Forecasts with high skill have great economic value for a multitude of sectors and many millions of dollars could be saved by further improving forecasts (Katz and Murphy, 1997; Nurmi et al., 2012). Minimizing the false-positives and false-negatives of critical weather events is essential to the productivity, as well as safety, of much of the public. Stakeholders often take action in response to a forecast probability of surpassing a 'critical threshold' (Richardson, 2000). The forecast is useful if the stakeholder benefits from this action (Richardson, 2000; Chatrchyan et al., 2017). For example, in the agricultural

sector, proper forecasts of when subfreezing temperatures occur are needed so that managers can treat their crops accordingly. However, false-positive identification of freeze events will result in unneeded crop protection and thus an economic loss (Lave, 1963; Katz et al., 1982). Fitting a Gaussian PDF to a non-Gaussian distribution such as that illustrated in Fig. 1(d) can assign very different probabilities to surpassing a specific threshold.

Identifying systematic bimodality in ensemble forecasts may also provide a means of studying weather 'regimes' (Birchfield

et al., 1990; Morrison et al., 2012; Fallah and Sodoudi, 2015). Michelangeli et al. (1995) described weather regimes as states of the atmosphere that exhibit the properties of recurrence, persistence and are quasi-stationary. One reason for systematic bimodality may be that different members of the forecast ensemble enter and reside in distinct regimes; thus identifying this bimodality may provide a new way of identifying and studying these regimes. For instance, the bimodal forecast exhibited in Fig. 1(d) is for a location in the Southern Ocean. The warmer of the two modes lies close to the freezing point of water,

suggesting that in this state there may be no sea-ice, while in the colder mode sea-ice could be present, insulating the air from the warmer ocean below. The likelihood and accurate representation of the possibility of each of these scenarios is vital and has important implications for studying ocean-atmosphere interactions (Zippel and Thomson, 2016).

Finally, accounting for bimodality in the error statistics of the forecast also has technical implications for the design of the forecast and data assimilation system (Dovera and Della Rossa, 2011; Miller and Ehret, 2002), and in the assessment of the

relationship between forecast spread and the skill of the forecast system (Wilks, 2002). The identification of specific weather regimes may allow for better bias corrections in forecasts that align with these regimes (Allen et al., 2019), leading to improved forecast scores and better understanding of model biases.

This paper is organized as follows: Section 2 presents a statistical test for bimodality. In Sect. 3 these methods are applied to an ECMWF dataset and results are presented. Discussion of these results takes place in Sect. 4. Section 5 explores and illustrates

difficulties and deficiencies of current skill-scoring methods when applied to non-Gaussian distributions; these results indicate the need for revised ensemble scoring methods to assess and account for bimodality. Finally, Sect. 6 concludes.

## 2 Ensemble Fitting and Defining Bimodality

There are several ways ensemble forecasts, which are made up of discrete points, can be 'dressed', meaning a continuous PDF is fit to the ensemble. As mentioned previously, a Gaussian PDF using the mean and standard deviation of the ensemble is quite common, especially due to its computational efficiency. However, this proves to be insufficient for the current study due to its unimodal constraint. For this study, a non-Gaussian dressing method, the kernel density estimate, is used. Explicitly, this is a Gaussian kernel density estimate (KDE) (Silverman, 1981) whose resulting PDF can be written

$$P_{KDE}(T) = \frac{1}{\sqrt{2\pi}nh} \sum_{i=1}^{n} \exp^{-\frac{1}{2}\left(\frac{\bar{T}-T_i}{h}\right)^2} \tag{1}$$

where $n$ represents the number of members in the ensemble and $h$ represents the bandwidth, a smoothing parameter. $h$ was determined using Scott's rule (Scott, 2015). The relative minima ($T_{m,i}$) and maxima ($T_{M,i}$) of this distribution are identified by applying a root-finding method to the first derivative of the PDF.

The main requirement to detect bimodality is of course the presence of two relative maxima. However, the presence of bimodality in an estimated PDF may be a result of sampling error due to a finite ensemble size. To reduce the frequency of such false positives while also minimizing false-negatives in the presence of bimodality, two other criteria were also required to be met before a forecast was determined to be bimodal. The first criterion involves the 'mode probability ratios', defined as the ratios of the probability of the two local maxima $P_{KDE}(T = T_{M,i})$, relative to the probability of the local minimum $P_{KDE}(T = T_{m,i})$. Values of the mode probability ratios for two example multimodal distributions shown in Fig. 2(a) are indicated in the legend. Note that, for a bimodal case, this ratio depends on which mode in the distribution you measure against.

The second criterion involves the membership of each mode. To define which ensemble members belong to which mode in a multimodal distribution, the relative minimum is used as a cutoff point. Using a bimodal distribution as an example: an ensemble member belongs to mode 1 if the temperature is in the range $(-\infty, T_{m,1})$; it belongs to mode 2 if it was in the range $(T_{m,1}, +\infty)$. This is illustrated in Fig. 2(b) with two modes separated by the minimum location (black dotted line). Members belonging to mode 1 are indicated in blue and those belonging to mode 2 are in red.

### 2.1 Selection of Bimodal Criteria

To choose appropriate thresholds for these two criteria, a series of tests were applied to synthetic 50-member forecasts drawn from a Gaussian mixture model

$$P_{GM}(T) = \frac{1}{2}\mathcal{N}(T; -\delta/2, 1) + \frac{1}{2}\mathcal{N}(T; \delta/2, 1), \tag{2}$$

with increasing values of $\delta$. Here $\mathcal{N}(T; \mu, \sigma^2)$ is a Gaussian distribution with mean $\mu$ and variance $\sigma^2$. Note that values of $\delta$ from 0 to about 2 result in an underlying distribution that is unimodal; the true modes of this distribution are indicated in Fig. 3(a).

A KDE was applied to each synthetic forecast, and the relative extrema were found. To be identified as bimodal, the KDE must have two modes, the mode probability ratio must meet a minimum threshold, and each mode must contain a minimum

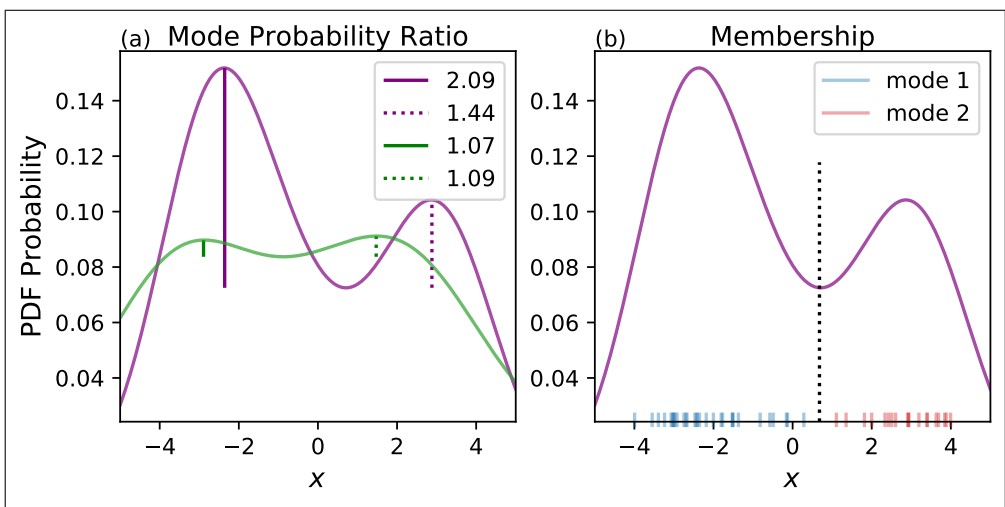

**Figure 2.** (a) Synthetic data representing how the magnitude of the mode probability ratio has been quantified for two distributions. The mode probability ratio will vary depending on which mode you are measuring against. (b) How ensemble members have been assigned to a mode based on if they are less than the local minimum value in the PDF of the distribution (blue) or greater than the local minimum (red).

number of members. As a base case, the two additional criteria were omitted. This case proved to have a relatively large false-positive percentage. By including the two additional criteria, the false-positive likelihood was reduced.

In order to determine the appropriate combination of restrictions, the requirement for a minimum number of members, $m$, in the smaller of the two modes was first added. Unless the distribution satisfied this additional requirement, it was not considered 'bimodal'. Values for $m$ ranging from 1 to 25 were considered. This process allows one to derive false-positive and false-negative occurrences associated with each membership restriction. Requiring a minimum of 5 members in each mode was found to greatly reduce the false-positive rate, while also keeping the false-negative rate low (not shown).

The 5-member requirement was then combined with minimum mode probability ratio requirements ranging from 1 to 10. There are two possible ratios for each minimum depending on whether the local maximum to the left or to the right is chosen. In order to be considered bimodal, three possibilities were considered: *either* ratio must exceed the requirement, *both* ratios must exceed the requirement, or the *average* of the two ratios must exceed the requirement.

An example of these tests is illustrated in Fig. 3(b). This depicts the false-positive and false-negative percentages for a 5-member minimum in combination with the requirement that *either* mode probability ratio exceed the specified value. Three different underlying distributions drawn from Fig. 3(a) are represented– a distribution on the cusp of unimodal and bimodal (solid blue), a bimodal distribution with a relatively large $\delta$ (blue dash-dot), and a distribution in the middle (blue dashed). The false-positive rate associated with a normal Gaussian distribution is also indicated (black dotted).

Based on these tests, the following requirements were adopted for identifying a distribution as bimodal: the sample must have two relative maxima separated by one relative minimum, each mode must contain at least 5 ensemble members and at least one of the mode probability ratios must exceed a value of 1.18. A mode probability ratio of 1.18 represents a minimum that

has a probably at most of 85% of one of the maxima. The mode probability ratio of 1.18 is highlighted in Fig. 3(b). This value was chosen because it is near the inflection point of the false-positive curve, reducing incorrect detection of bimodality, while also staying to the left of the false-negative curves, where missed detections quickly rise. How well the combination of these three criteria perform on predicting the modality of all the distributions in Fig. 3(a) can be found in blue in Fig. 3(c). These metrics are quantified from 10,000 randomly drawn 50-member ensembles. The vertical blue dashed line represents when the distribution switches from unimodal to bimodal. It can be seen that for values of $\delta$ less than about 1, this test identifies nearly 95% of the forecasts as unimodal, corresponding to a false-positive rate of 5% (blue solid). This is well improved from a test that only has a two-mode requirement which results in a false-positive rate of 20% (blue dotted). For values of $\delta$ greater than about 4, the three-criteria-test identifies nearly all forecasts as bimodal. Distributions on the cusp of bimodality for which $\delta$ lies between 2 to 4 present a greater difficulty.

Two other distributions were tested in a similar fashion:

$$P_{GM}(T) = \frac{3}{4}\mathcal{N}(T; -\delta/2, 1) + \frac{1}{4}\mathcal{N}(T; \delta/2, 1) \tag{3}$$

which is shown in red in Fig. 3(c), and

$$P_{GM}(T) = \frac{3}{4}\mathcal{N}(T; -\delta/2, 2) + \frac{1}{4}\mathcal{N}(T; \delta/2, 1) \tag{4}$$

which is shown in green. The distribution represented by (3) becomes bimodal near $\delta \approx 4$, while (4) becomes bimodal near $\delta \approx 5.5$. These two additional distributions are used to determine the robustness of these restrictions to varying degrees of skew. Figure 3(c) shows that for $\delta$ less than about 2, these distributions also have false-positive identification rates of about 5%.

Correctly identifying the presence of bimodality for values of $\delta$ near where the underlying distribution transitions from unimodal to bimodal is more challenging, particularly for the greater skewed distributions. In this portion of parameter space a greater proportion of false-positives and false-negatives would be expected.

## 2.2 Application to a More Sophisticated System

The analysis of these criteria may be expanded upon by testing the false-positive and false-negative rate on a more sophisticated system. The Lorenz 1963 model allows for the representation of a chaotic system with three degrees of freedom (Lorenz, 1963). This model has often been used as a simplistic representation of the atmosphere. For this study of bimodal detection, the system is initialized with a very large ensemble (10,000 members) where members are perturbed relative to one another. Since the flow is chaotic, the model is sensitive to initial conditions and thus the ensemble members will spread. This large ensemble may be used as the 'true' distribution for which the modality of the 'atmosphere' at that point in time may be found. Then, similar to the previous synthetic study, 50-member ensembles at discrete timesteps can be drawn, where each are fit with a KDE and the derived bimodal criteria are applied to get an approximation of the false-positive and false-negative detection rate. Figure 4 depicts the time evolution of a univariate variable in a Lorenz 63 system. This example, and more generally most ensemble forecasts of the atmosphere, may be broken into three main sections or periods: 1) just after initialization when the ensemble is still tightly clustered around the 'true' state (roughly lead times 0-5 in the case of Fig. 4), 2) after the ensemble has begun to

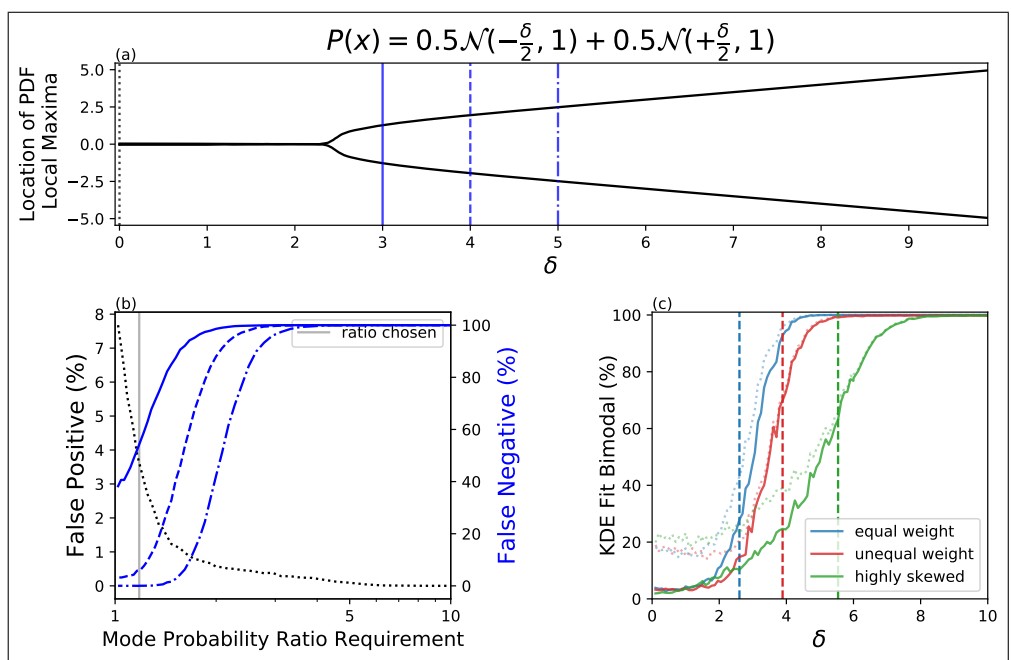

**Figure 3.** (a) The local maximum location(s) for a series of bimodal distributions with increasing $\delta$ according to the PDF of that distribution. (b) The false-positive (black dotted) and false-negative (blue) rate that is associated with various minimum mode probability ratio requirements in order to classify the bimodality of a sub sampled 50-member sample fit with a KDE for three different underlying distributions (solid, dotted, dot-dashed) (the mode probability requirement is in combination with a 5-member minimum mode requirement). The mode probability requirement used in (c) is demarcated by a vertical gray line. (c) The bimodal fit percentage for a 50-member sample fit with a KDE. Vertical dashed line represents when the underlying distribution truly transitions from unimodal to bimodal. The correspondingly colored dotted line represents the percentage of time the KDE classified bimodal with only a two-mode requirement. The correspondingly colored solid line represents the same statistic except while using the restrictions of: two modes, either mode probability ratio must be greater than 1.18, and a 5-member minimum for each mode. The blue distribution represents the distribution in (a). The red distribution represents the same distribution as (a) but with a weight of the lower mode 3 times that of the higher mode; for many values of $\delta$ this would be representative of a skewed distribution. The green distribution represents the same as (b) but the greater weighted mode has a standard deviation of 2 instead of 1; for many values of $\delta$ this results in a highly skewed distribution.

spread, where non-Gaussian characteristics may or may not be present (leads 5-25 in Fig. 4) and 3) at long lead times when the ensemble approaches the climatological distribution (leads 25+ in Fig. 4). While bimodality can be present in the climatological distribution itself, the bimodality that this manuscript is primarily concerned with identifying would be expected to occur in the second period and not necessarily persist into the third. Figure 4(b) has a sample distribution and the evaluated false-positive and false-negative rate from each of these three periods. These rates correspond well to the previous synthetic study. In this

example, there is a larger false-positive rate associated with the 'climatological' distribution, which has greater spread than the initialization period. Note that this example also reflects how although a forecast may experience a brief period of bimodality

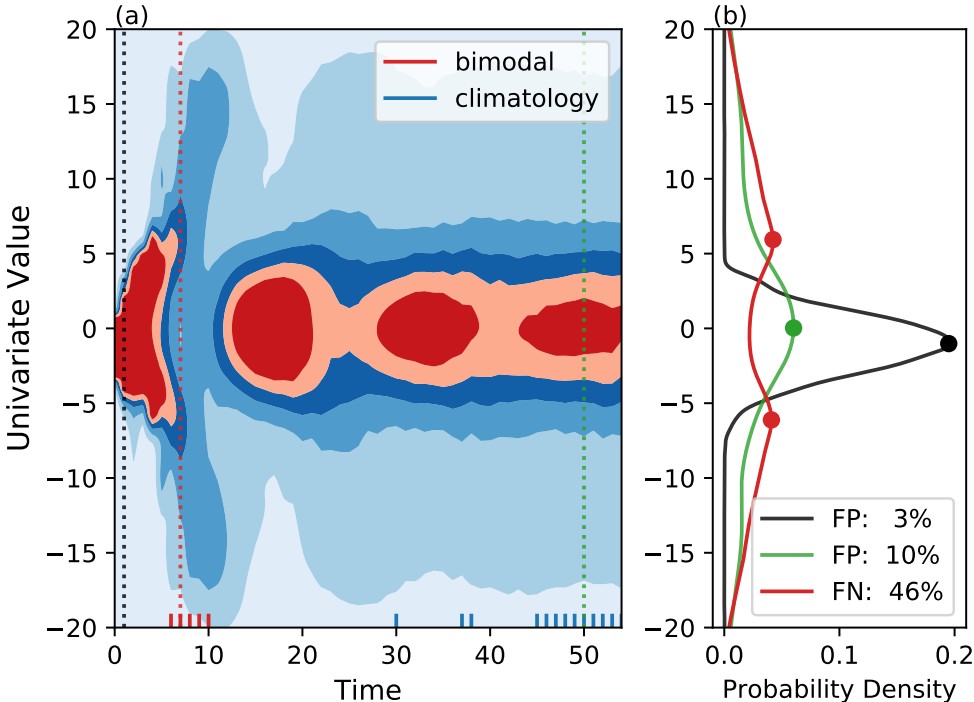

**Figure 4.** (a) The time evolution of a 10,000 member ensemble in a Lorenz 1963 system. $x_o = (-4.11, -1.29, 26.37)$, $r = 28$, $p = 10$, $b = 8/3$. Shading indicates the kernel density estimate (KDE) of the univariate ensemble distribution. Darker shades of red indicate a high number of ensemble members near that value. Red tick marks on the x-axis indicate timesteps when the ensemble has two modes. Blue tick marks indicate those distributions who are significantly similar to the 'climatological' distribution according to a two-sided Kolmogorov-Smirnov test ($\alpha = 0.025$). Climatology was determined by integrating the model until $t = 200$. (b) Discrete distributions drawn from the timeseries in (a). Dots indicate where the maxima occur. The legend indicates the false-positive (negative) detection rate of unimodal (bimodal) distributions according to repeatedly drawn 50-member ensembles with the bimodal criteria derived from Fig. 3 applied.

(leads 6-10), it does not necessarily mean that this bimodality will persist for the entirety of the second characterized 'period' (leads 10-25).

On the basis of Fig. 3 and Fig. 4, if this test identifies bimodality in approximately more than 5% of forecasts, one can
safely conclude that non-Gaussian distributions are systematically present. For distributions that are sufficiently skewed, there is a greater potential for false-positives and false-negatives, especially near the transition between unimodality and bimodality. However, it is difficult to quantify this potential for the forecast system of interest without knowing beforehand the appropriate family of possible distributions that the forecasts may exhibit. Nonetheless, as seen in a later section, estimates of $\delta$ in forecasts that are found to be bimodal with this test are often well above the transition region indicated in Fig. 3(c).

## 2.3 Transient Versus Climatological Bimodality

Before this procedure is applied, however, one may wish to understand to what extent the bimodality that is captured could be a result of the climatological distribution itself (period three from Fig. 4) versus a specific atmospheric state at the time of the forecast (period two from Fig. 4). If the occurrence of bimodality is a result of the climatological distribution itself, one would expect occurrence rates to asymptote at very long forecast lead times as the forecasts approach the climatological distribution. However, if bimodality is occurring due to particular atmospheric states, one would expect there to be some 'peak' in bimodal frequency at some function of lead time prior to an asymptotic behavior, this 'peak' period would be representative of the second period in Fig. 4.

Using the forecast location from Fig. 1(d), it is found that even at the longest lead times offered by the ECMWF dataset (weeks 4+) the forecast distributions at this location are still significantly different from that of climatology (according to a two-sided Kolmogorov-Smirnov test with $\alpha = 0.025$). This indicates that there may be some low-frequency variability that could contribute to potential forecast skill even at these very long lead times. This also means that the ensemble forecast bimodality that is expressed (for example in Fig. 1(d)) is at least partially due to a more transient process rather than just a sampling of the climatology itself. More work has gone into uncovering this on a broader scale and similar conclusions have been found, however, the complete presentation of this work is beyond the objectives of the current manuscript.

## 3 Bimodality Statistics

The detection and quantification of bimodality, as derived from the tests defined in Sect. 2, was applied to real forecast data. The dataset used is the ECMWF Atmospheric Model Ensemble extended forecast (ENS extended). This dataset includes 46-day, 50-member forecasts every Monday and Thursday (Haiden et al., 2019). Forecasts extend from 03 December 2015 until 28 January 2021. This study will limit its analysis to the identification of bimodality in 2-meter temperature. Furthermore, the analysis will be focused on three main factors: the occurrence of bimodal forecasts, the separation between the two modes of bimodal forecasts ($\delta$) and the greater of the two mode probability ratios of bimodal forecasts.

Figure 5 shows the occurrence rate of bimodality as a function of the lead time of the forecast. Three different sets of forecast leads are considered based on what lead time they are valid for: week one, weeks two and three, and weeks four and five. The color scale is set so that any shaded region indicates more than 5% of forecasts at these lead times exhibit bimodality. The darkest shades of blue represent 30% or more of forecasts that are bimodal. At all three time periods, values of bimodal occurrence for much of the globe are well over the expected false-positive rate of 5% determined from synthetic testing; this suggests that bimodality is systematic and widespread in forecasts of 2-meter temperature.

Generally, the occurrence of bimodality in forecasts increases from week one to weeks two and three. Interestingly, though, the occurrences also become more localized; there are almost no locations that are below the 5% threshold in the first week (Fig. 5(a)), however, this occurs more widely during weeks two and beyond (Fig. 5(b,c)), especially over the eastern parts of extratropical ocean basins. At weeks four and five, the global occurrence rate decreases from weeks two and three and the spatial area of those regions that experience less than 5% bimodality grows larger. These properties are expressed in the global

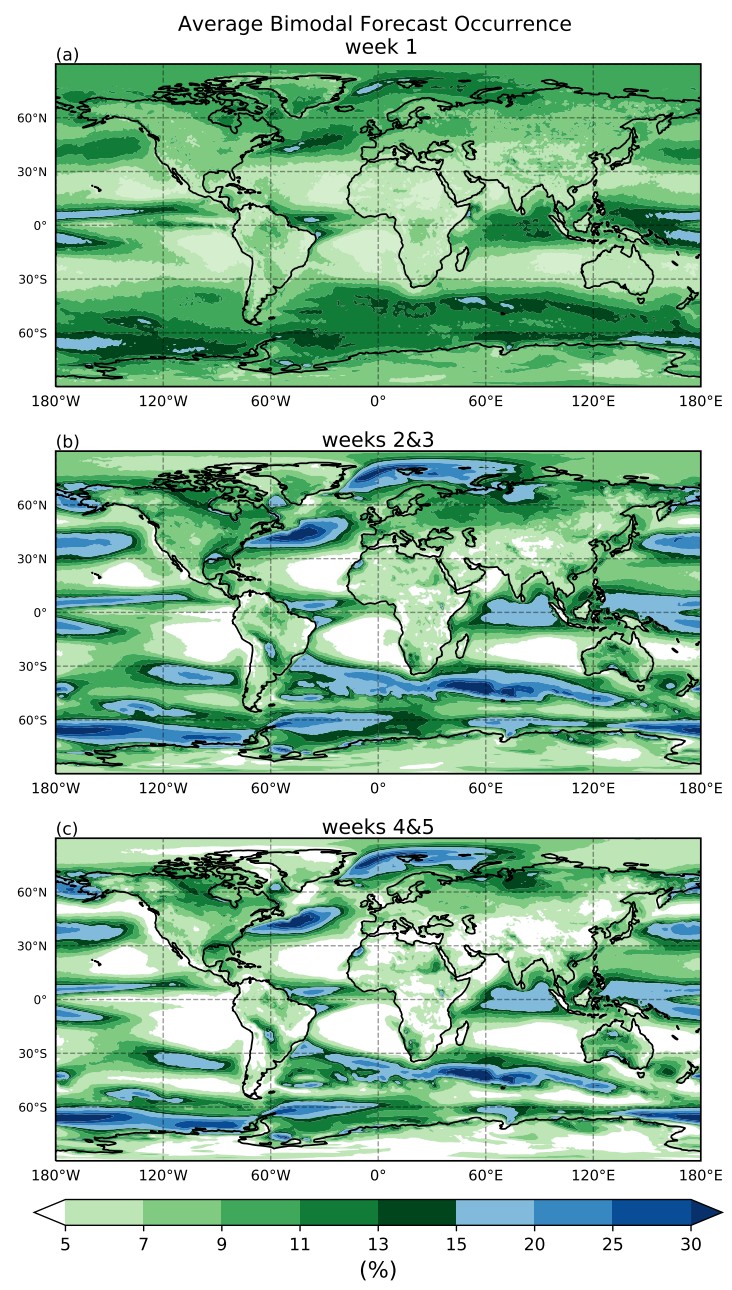

**Figure 5.** (a) The average occurrence of forecasts that are considered bimodal in the first week of lead times, (b) in weeks 2-3, (c) in weeks 4-5 for the ECMWF 2-meter temperature dataset.

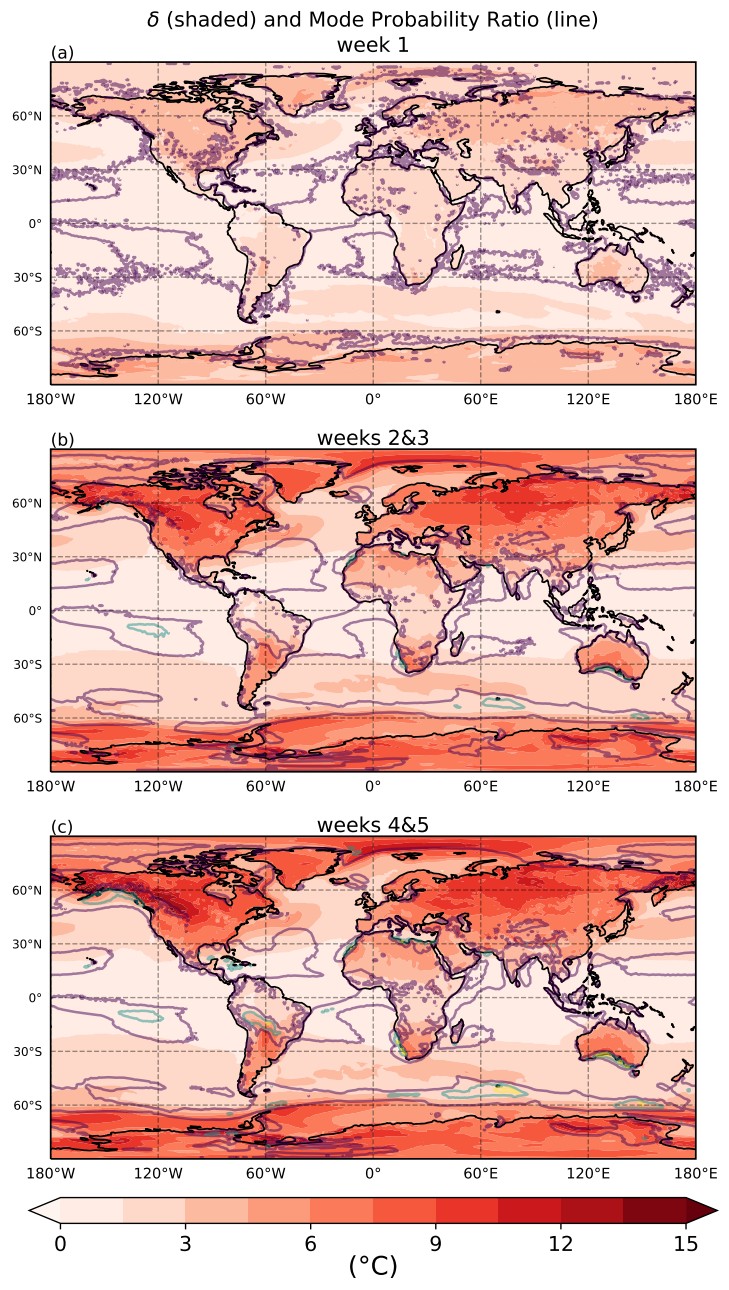

**Figure 6.** Shading represents the average $\delta$ between the two modes of forecasts deemed bimodal. Darker shades of red indicate greater separation of temperature between the two modes. Contours represent the mode probability ratio of the absolute maximum compared to the relative minimum of forecasts deemed bimodal. Purple, blue and yellow contours represent 2 times, 3 times, and 4 times as likely, respectively. (a) is for lead times in week 1, (b) in weeks 2-3, and (c) in weeks 4-5 of the ECMWF 2-meter temperature dataset.

mean and standard deviation of the bimodal forecast occurrence as well. Week one of forecasts has a mean occurrence of 9.1% with a standard deviation of 2.6%. Weeks two and three of forecasts have a mean occurrence of 10.2% with a standard deviation of 5.2%. Weeks four and five have a mean occurrence of 8.9% with a standard deviation of 5.2%. With the exception of the eastern ocean boundaries, the occurrence of bimodal forecasts is generally larger over the oceans as compared to land.

Figure 6 shows the evolution of $\delta$ and the mode probability ratio as a function of lead time. The $\delta$ between bimodal distributions' modes generally increases as a function of lead time. $\delta$ is generally larger over the land versus the ocean. An interesting property that develops in the mode probability ratio is the decreased spatial extent of the lower probability ratios after the first week of lead times (represented by the decreased occurrence of purple contours in Fig. 6(b) as compared to Fig. 6(a)). Probability differences become greater in magnitude but more localized after the first week of lead times. $\delta$ values and local maxima in mode probability ratio continue to grow into weeks four and five of lead times. It is important to note that the $\delta$ and mode probability ratios are averaged only over those forecasts which are considered bimodal. Thus, the localization pattern just expressed may be a result of smaller sample sizes associated with the first week of forecasts (Fig. 5(a)), and perhaps not from the properties of the forecasts themselves. An example of where this may be the case is in the Southern Ocean near (60$^o$ S, 60$^o$ E).

Next the seasonality of bimodality is analyzed. Only lead times from two to three weeks are focused on since this period exhibits the greatest occurrence of bimodality (Fig. 5(b)). Nonetheless, the other lead times represented in Fig. 5 exhibit very similar properties in seasonality.

In Fig. 7 and Fig. 8 the same information from Fig. 5(b) and Fig. 6(b) is being shown, but now only for those forecasts initialized during Northern Hemisphere extended winter (November through March) (Fig. 7) or Northern Hemisphere extended summer (May through September) (Fig. 8). There is clear evidence of increased bimodality in forecasts that occur in the winter hemisphere, represented by darker greens and more blue regions in the Northern Hemisphere in Fig. 7(a) and in the Southern Hemisphere in Fig. 8(a). As seen previously, greater bimodality occurs over the ocean as compared to over land in both hemispheres for both the warm and cold season. However, there are localized patches of bimodality maxima that reach up to 30% over some continental regions such as near Morocco and east of the Andes during MJJAS. Examples of high land occurrences during NDJFM include Australia, Canada, western Alaska, and the southeastern U.S. Regions near the equator appear to have relatively high occurrences in bimodality during the warm and cold seasons. Conversely, bimodality minima persist through both extended summers and winters in the midlatitudes off the western edge of most of the continents.

Although values of $\delta$ are typically largest over the land as compared to the ocean, there is a stronger dependence on season for forecasts over land versus those over the ocean. $\delta$ values are generally largest over land in the winter hemisphere, but are somewhat comparable no matter the season over the oceans outside of the polar regions. In many areas, average $\delta$ values of over 15 $^o$C are evident in land forecasts that are bimodal. This includes much of Canada, regions east of Greenland, much of northern Russia and some of Australia during boreal winter. Forecasts with large $\delta$ values during boreal summer include central South America and almost the entirety of Antarctica. Sea ice regions in both hemispheres exhibit especially large values of $\delta$. As with the nonseasonal pattern (Fig. 6), values of $\delta$ continue to increase into weeks four and five of forecast lead times.

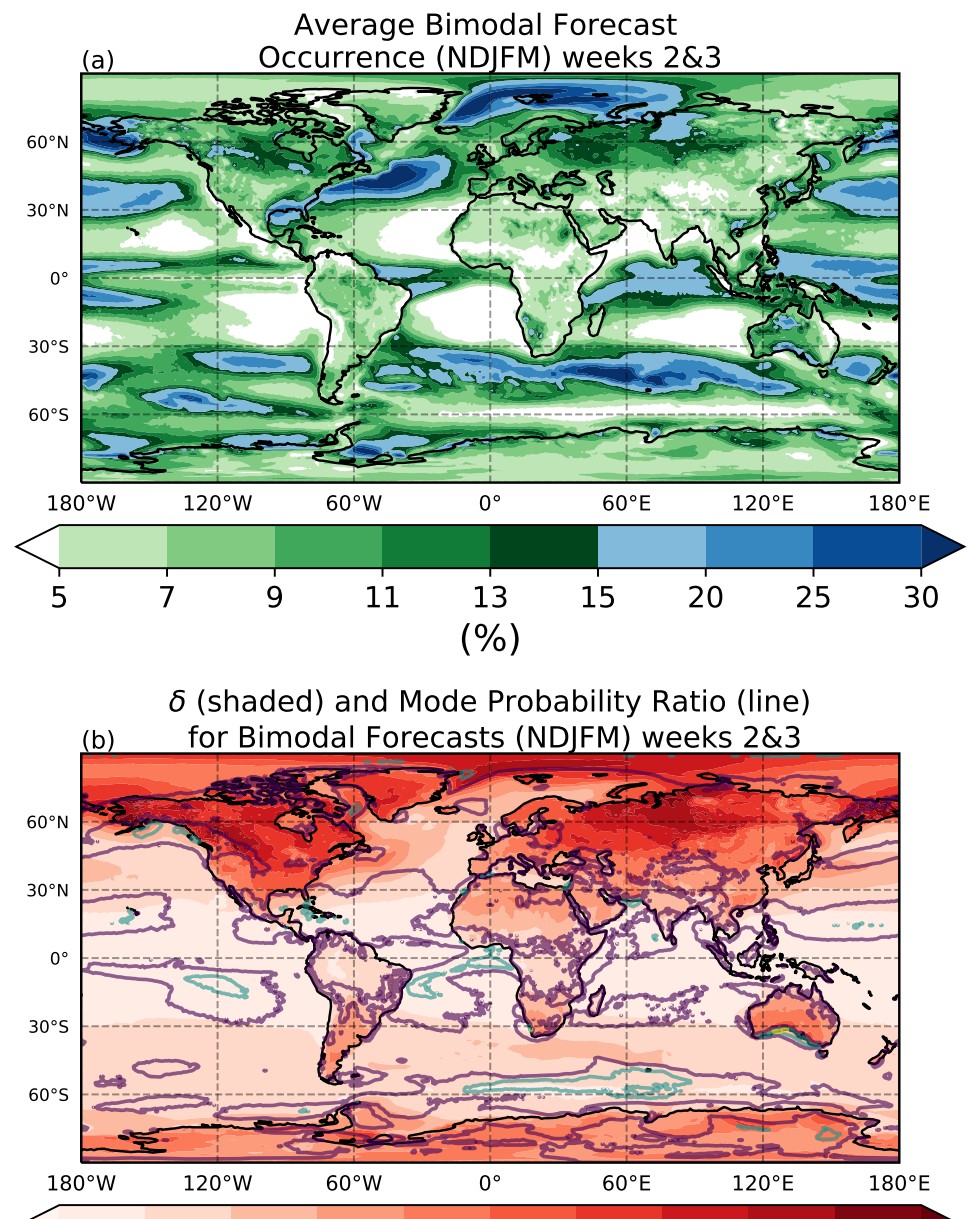

**Figure 7.** (a) The average occurrence of forecasts that are considered bimodal for all Northern Hemisphere extended winter months (November, December, January, February, March) in the ECMWF dataset. (b) The accompanying average delta between the two modes (shaded) and the relative probability of the absolute maximum compared to the relative minimum for forecasts that are bimodal (contour). Purple, blue and yellow contours represent 2 times, 3 times, and 4 times as likely, respectively.

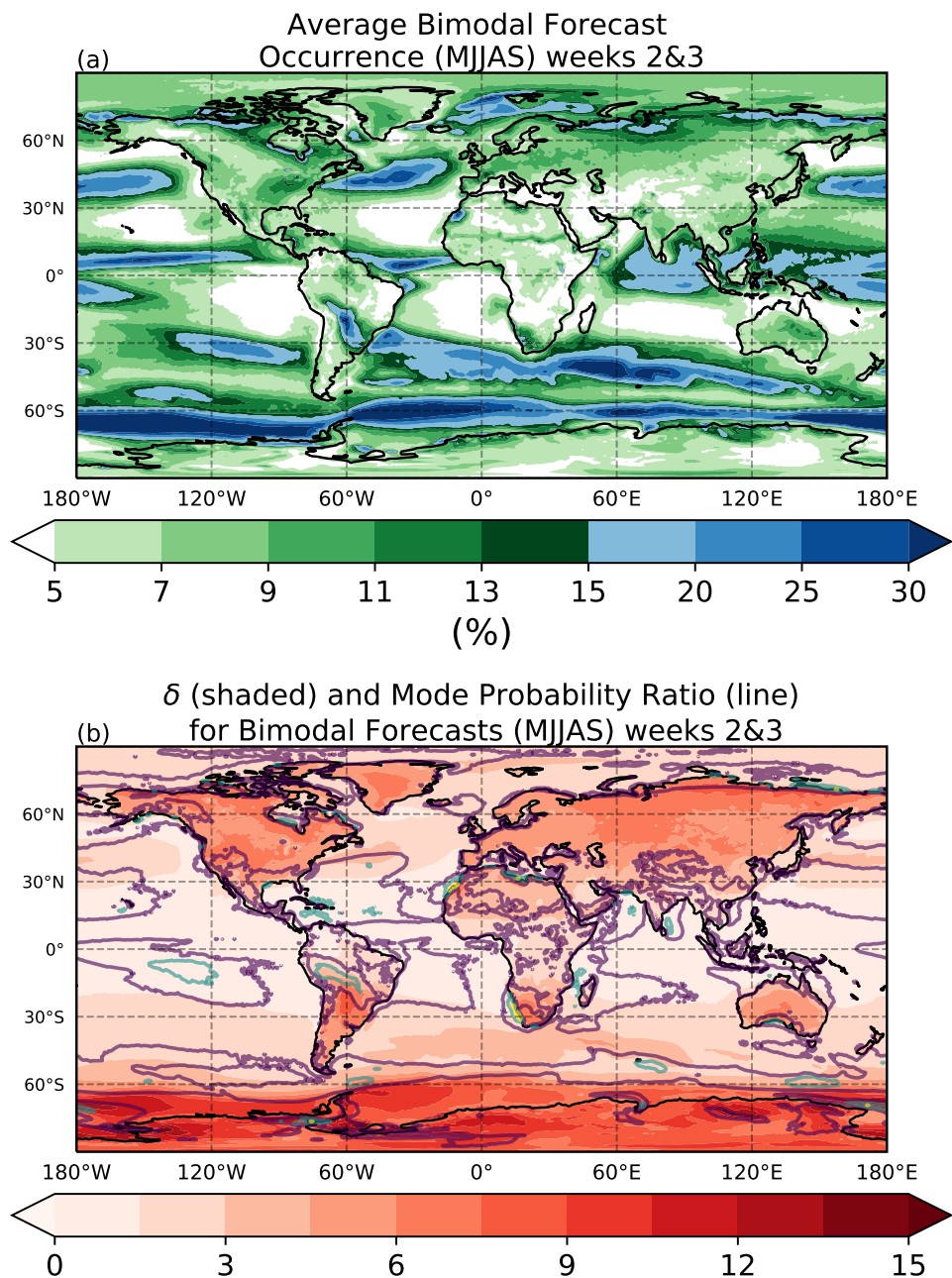

**Figure 8.** (a) The average occurrence of forecasts that are considered bimodal for all Northern Hemisphere extended summer months (May, June, July, August, September) in the ECMWF dataset. (b) The accompanying average $\delta$ between the two modes (shaded) and the relative probability of the absolute maximum compared to the relative minimum for forecasts that are bimodal (contour). Purple, blue and yellow contours represent 2 times, 3 times, and 4 times as likely, respectively.

Unlike occurrence and $\delta$, the mode probability ratios do not exhibit as pronounced of a seasonal dependence. There is some variation in the maxima that occur in the Southern Ocean, southern Australia, and west coast of Africa which are greater in austral summer than in austral winter. However, this may be affected to some extent by seasonal changes in the occurrence frequency. Regardless of season, there are persistent maxima to the west of South America, near the Gulf of Mexico, and southern Australia.

Finally, note in particular those locations that experience both large $\delta$ as well as large mode probability ratios. These areas represent not only forecasts that have very widely spaced modes, but also with PDFs that may contain very distinctly separated modes, or relative minima that are very near a probability of 0. From properties presented previously (Fig. 1(d)), this minima will commonly be near where a Gaussian fits the greatest probability. Additionally, since the $\delta$ is so large, the Gaussian will have a very large variance. Hence in these regions assumptions of Gaussianity are the most problematic. Examples of these locations include the southern coast of Australia and western Alaska during boreal winter.

## 4  Discussion

With only a local, univariate analysis, it is difficult to determine the cause of the patterns reported in the previous section. However, based on the regional patterns and structures some hypotheses can be made.

The increase in the occurrence rate of bimodality beyond lead times of one week is consistent with the idea that non-linear spread requires time to develop. Given the tendency for these forecasts to be underdispersed at these lead times (e.g. Fig. 1(a)), it is all the more remarkable that there are still indications of bimodality at these early lead times. Additionally, because the occurrence peaks during weeks two and three of the forecasts and then decreases, it is unlikely that the bimodality in these forecasts are simply a result of the approach to a climatological distribution. If this were the case, one would expect to see the occurrence frequency asymptote at longer lead times.

In both hemispheres, bimodality occurs more frequently over the ocean than over land. This suggests an important role for boundary layer processes in coupling the atmosphere to the ocean (Chelton, 2013). This is especially evident in the Gulf Stream area off the northeast coast of the United States, the Southern Indian Ocean, as well as in polar regions off the eastern coast of Greenland and in the Southern Ocean. However, there seems to be a greater seasonal dependence in bimodality occurrence for these polar regions as compared to the Gulf Stream and the Indian Ocean regions. This further suggests a role for sea ice in the polar regions where bimodality occurs in nearly 30% of forecasts, with average values of $\delta$ of over 15 $^o$C. The high occurrence rates of bimodality in the Gulf Stream or regions with sea ice may be due to uncertainty associated with advection of surface winds over strong ocean temperature gradients. Whether the ensemble members predicts surface winds pointing from the warm to cold region or vice versa has wildly different outcomes on the 2-meter temperature.

More generally, one reason for the seasonal dependence on bimodality occurrence may be the presence of stronger meridional temperature gradients in the winter hemisphere. These gradients lead dynamically to increased baroclinic instability and the effects of advection around these gradients can produce strong non-Gaussianity. In a simple case, take a series of ensemble members' whose initialization result in a similarly structured cyclone that produces warm air advection to the east of the cy-

clone. In contrast, the rest of the members do not predict the formation of a cyclone and no anomalous advection takes place. This will result in the development of two modes in the forecast distribution, one unperturbed mode and one anomalously warm mode. Now consider that in the winter hemisphere these cyclones can develop faster, become stronger, as well as have a greater magnitude in anomalous temperature advection for the region. This may act to more quickly and dramatically separate modes in the forecast distribution. This greater separation would not only lead to higher occurrence of bimodality in the winter hemisphere but also greater values of $\delta$, both of which are supported by the patterns seen in Fig. 7 and Fig. 8. Tamarin-Brodsky et al. (2019) depicted climatological skewness that develops in ensemble forecasts of temperature near storm track regions as a result of nonlinear meridional advection. Consistent separation of smaller modes in forecasts, such as those presented in the previous hypothetical case, may express themselves in the climatological distribution as skewness, which would be consistent with the findings from Tamarin-Brodsky et al.. That being said, there are clear distinctions between the spatial distribution of temperature skewness found in Tamarin-Brodsky et al. (2019, their Fig. 1(c)) and the distribution of bimodality exhibited in the current study. Examples of uniqueness for boreal winter include west of South America (a minima in bimodality but large skew) east of Greenland (maxima in bimodality but little skew) and west of Australia (minima in bimodality but large skew). This indicates that while the processes which lead to each may be related, there are some distinctions. The relationships between increased baroclinicity, its effects on weather events, and the associated forecast uncertainty in the form of ensemble spread and non-Gaussianity remain a popular topic of study (Vallis, 1983; Scher and Messori, 2019; Linz et al., 2020).

Persistent bimodality minimia off the western edges of continents may be associated with high pressure systems typically sitting over these regions (Aguirre et al., 2019) which can suppress convective activity and reduce ensemble spread, mitigating non-Gaussianity in forecasts. However, it is interesting that the occurrence of bimodality in these regions is larger during the first week of forecasts than at longer lead times.

Smaller values of $\delta$ over the ocean as compared to continental regions are also consistent with the importance of air-sea interactions. The larger heat capacity of the ocean as compared to land surfaces is likely to reduce possible fluctuations in 2-meter temperature. This may also explain why there seems to be very little seasonality of $\delta$ over the ocean relative to the high seasonality of $\delta$ over land.

High bimodality occurrences surrounding the equator may be associated with convective activity near the ITCZ which may lead to non-Gaussianity in forecast distributions. The area of enhanced occurrence near the equator appears to shift slightly north of the equator during boreal summer and slightly south of the equator during boreal winter which is consistent with the positioning of the ITCZ. This is further reinforced by looking at a region such as the northern Bay of Bengal. This region switches from below 5% occurrences to nearly 20% depending on the season; however, this maximum in occurrence is achieved in the region's warm season, a property that is opposite of most of the globe. The seasonal location of this maximum, however, is consistent with the seasonal ITCZ position, where the ITCZ position is based off of rainfall climatology (Souza and Cavalcanti, 2009). Bimodality in this region during boreal summer may be also in part due to the timing of the monsoon circulation. When this circulation sets up or breaks down and whether ensemble members predict it or not will lead to largely diverging atmospheric states.

## 5 Ensemble Scoring

Bimodality clearly occurs quite frequently in these forecasts. One may now wish to understand what effect this has on forecasting skill when it is present. Do such cases lead to worse forecasts? Can the use of non-Gaussian dressing methods improve forecast skill relative to Gaussian approaches in these cases? How does bimodality affect postprocessing techniques that are also often based on assumptions of Gaussianity? This final section will be devoted to answering these types of questions

The use of postprocessing in ensemble forecasts can reduce systematic biases in the forecast model (Vannitsem et al., 2018).
This process involves scoring the ensemble against validating observations using some standard metric. On the basis of these scores, systematic corrections can be applied, such as a shifting the mean or dispersion of the forecast ensembles. The use of appropriate scoring methods is needed for this postprocessing to be successful. It is necessary to use a scoring metric which is 'proper' (Gneiting and Raftery, 2007) and which can assess metrics of interest in an ensemble distribution.

Two well known-scoring rules are compared: the continuous ranked probability score (CRPS) and the ignorance score
method (IS). In short, the CRPS is based on the cumulative density function (CDF) of the ensemble: the CRPS is defined as the squared difference in area above and below the CDF curve according to where the validation point lies. This is defined explicitly as

$$CRPS(F,x) = \int_{-\infty}^{\infty} (F(y) - H(y - x))^2 \, dy \tag{5}$$

where $F$ is the CDF of a probabilistic forecast, $H$ is the Heaviside step function and $x$ is the validation value. In contrast,
the IS is based on the PDF of the ensemble: it is defined as the negative log of the ensemble PDF at the value of the validating observation. Defined explicitly as

$$IS(P,x) = -\log P(x) \tag{6}$$

where $P$ is the probability density function of a probabilistic forecast and $x$ is once again the validation value.

For both quantities, low scores correspond to more skillful forecasts. Both scoring systems have usefulness such that they
can decompose a forecast system's reliability, resolution and uncertainty (Hersbach, 2000; Tödter and Ahrens, 2012). The ability to measure these metrics is attractive for those looking to formally evaluate forecasting skill. Extensive information can be found on both of these scoring methods (Vannitsem et al., 2018; Siegert et al., 2019).

### 5.1 Scoring Metrics' Ability to Resolve Modes

How these two scores assess ensemble forecasts of a synthetic bimodal process is considered in this section. First a 'perfect
model' Monte Carlo test is performed with an underlying bimodal distribution, representative of the flow state, from which 50-member synthetic ensembles are drawn. These forecasts are then dressed, which can then be used to compute the CRPS and IS. Two dressing methods are then compared, fitting either a Gaussian PDF using the mean and standard deviation of the ensemble, or using a Gaussian kernel density estimate as defined in Sect. 2.

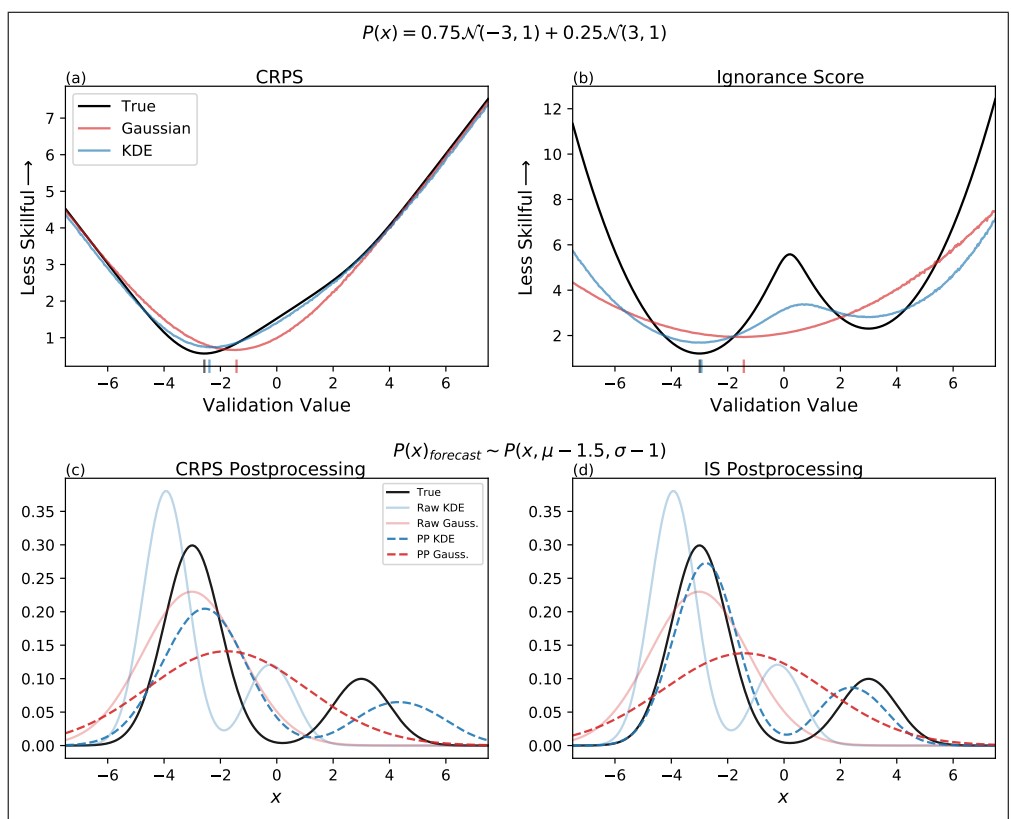

**Figure 9.** The comparison of two skill scoring methods, CRPS and IS, as well as Gaussian and KDE dressing methods. The underlying distribution that 50 member-samples were repeatedly drawn from is listed at the top. (a,b) The validation value is representative of what these 50 sized samples were scored against using the respective skill scoring method. Lower scores are representative of higher skilled ensembles. Ensembles dressed with a Gaussian distribution are in red. Ensembles dressed with a KDE are in blue. What the 'true' score values should be based on the underlying distribution is in black. Absolute minimum values for each distribution are demarcated with dashes on the x-axis. (c,d) How using CRPS and IS to postprocesses ensembles affects the ability of a biased forecast distribution to be corrected towards the 'true' distribution represented in (a,b). Postprocessed distributions are in dashed lines, raw forecast distributions are in solid lines.

The underlying true distribution is taken to be a mixed Gaussian, consisting of two unequally-weighted modes with unit standard deviation, centered at $\pm 3$. The mode at -3 has a weight of 0.75, while the mode at +3 has a weight of 0.25. In order to resolve the two modes, the skill score should reward forecasts if the validating observation lies close to the modes, while penalizing them if the validating observation lies at the minimum between the two modes.

Figures 9(a,b) show the CRPS and IS as a function of the validating observation for both of the dressed ensembles who each contain 50 members, averaged over 1000 forecasts. Also shown are the skill scores for a forecast distribution equal to the true distribution. The Gaussian distribution places the maximum weight at the mean of the ensemble, which tends to lie close to the mean of the underlying distribution, near the minimum in the true PDF between the two modes. Validating observations

near this point are given better scores than those that lie near the modes of the underlying distribution. The KDE distribution produces a better approximation to the underlying bimodal distribution, however, there is still no local minimum in CRPS resolved at 3 or a local maximum in skill score resolved at 0. In fact, should a validation of 3 occur, according to CRPS, the forecast will be scored worse than if a validation of 0 occurs, despite the mode at 3 in the true distribution. Even when using the true distribution, the only way in which the mode at 3 is reflected in the distribution of CRPS scores is through the rate of change in scores surrounding 3: scores increase (indicating worse skill) at a lower rate when transitioning from the mode at -3 to 3 as compared to validation values that lay beyond 3. Note that for outliers in the tails of the distribution, there is little difference in the scoring of the three forecast distributions.

The IS, in contrast, can more clearly resolve the presence of bimodality. The Gaussian dressed distribution again is scored most favorably if the validation observation lies close to the distribution mean. However, the KDE dressed distribution has local minima in skill score at both modes, with a maximum near the ensemble mean. The minima are even more clearly resolved if the true distribution is used; this is due to the known tendency for Scott's rule to overestimate the appropriate bandwidth for bimodal distributions. However, the outliers in the tails of the distribution are penalized much more severely for the KDE fit (or even moreso for the true distribution) than for the Gaussian fit. While all three distributions have Gaussian tails and thus will show quadratic growth in IS, the growth is much slower in the Gaussian case.

## 5.2 Scoring Metrics' Ability for Bias Correction

The consequences of these scores on the postprocessing and scoring of imperfect forecasts can further be explored. Here an underlying forecast distribution that is biased relative to the true process is considered: a mixed Gaussian distribution with the same distribution represented in Fig. 9(a,b) but with each mode shifted to the left by 1.5 units, and with a reduced standard deviation of 1 unit. The forecast and true distributions are shown in faint solid lines in Fig. 9(c,d). Then 2000 sets of 50-member ensembles are drawn from the forecast distribution and 2000 validation observations from the true distribution are drawn; from these an optimal calibration to the mean and dispersion is computed based on each scoring and fitting method. Explicitly, this involves optimizing one parameter which shifts the mean as well as one parameter which shifts the dispersion, where the optimization minimizes the fit distribution's skill score. This results in 4 sets of postprocessed distributions: a Gaussian and KDE fit optimized with CRPS (dashed red and blue lines, respectively, in Fig. 9(c)) and a Gaussian and KDE fit optimized with IS (dashed red and blue lines, respectively, in Fig. 9(d)).

From the distributions, it can be seen that since CRPS fails to resolve the second peak but instead has a slower dropoff in skill between 0-3 than to the left of -3 or right of +3, both postprocessed distributions are shifted to the right. The mean of the Gaussian distribution moves from the true distribution's largest mode to near the true distribution's local minimum as a result of the postprocessing. After postprocessing, the KDE's left and right modes are biased and skewed positive. Both distributions become overdispersed using CRPS for bias correction as compared to what the dispersion is of the true distribution.

In contrast, using the IS to postprocess the distributions gives better results. Due to the unimodal nature of the Gaussian fit, the mean is once again placed near the true distribution's local minimum, giving a similar result to the CRPS case. However,

the postprocessed KDE distribution approaches very nearly the true distribution. Note, however, that the degree of agreement is somewhat sensitive to the exact nature of the bias in the raw forecast distributions.

## 5.3 Number of Forecasts Required to Approach 'True' Distribution

Next the efficiency for each dressing method to reconstruct the forecast PDF and how close it comes to the scoring of the 'true' distribution is evaluated. The 'True' column in Table 1 shows the 'perfect' score for the two different skill score metrics, where this score is evaluated using the true distribution (black from Fig. 9). The 'Gaussian' and 'KDE' columns in Table 1 show the average scores (of 50,000 draws) for postprocessed distributions fit with each of the respective dressing methods for both evaluation and for the postprocessing procedure. For example, the 'Gaussian' column for the 'CRPS' row is showing what the mean evaluated skill score is for a Guassian dressed ensemble that is corrected by a postprocessing procedure, where the postprocessing minimized the CRPS of a Gaussian dressed ensemble. For the present bimodal distributions, the KDE performs better than the Gaussian fit for both scores, achieving an average score closer to that obtained if the true distribution is used. However, the improvement in the case of CRPS is very slight and requires a large sample to detect. This is illustrated by the 'No. required' column, which shows the number of 100-set 50-member forecasts required to score before the KDE and Gaussian scores can be distinguished at a confidence level of 95%. Explicitly this is testing the required number of forecasts, for each of the 100 sets, before the two fitting methods are significantly different based on the variance of the mean of the 100-set forecasts. More than 16,000 forecast-sets are required to demonstrate that the KDE is a better choice, despite the bimodality of the underlying distribution. In contrast, if IS is used, roughly 70 forecast-sets are required to demonstrate that the KDE is a better choice. Further tests indicate that the choice of score used to perform the postprocessing is more important than the score used to evaluate the forecasts (not shown).

These tests were also conducted with smaller ensembles to evaluate the effectiveness of the two skill scoring methods with ensembles of different sizes. Smaller ensembles dressed with KDEs did not perform as well and had a mean IS further away from the 'perfect' IS than larger ensembles fit with KDEs. The effect on IS for Gaussian fit ensembles was less noticeable, but still had scores further away from the 'perfect' score as ensemble sizes got smaller. CRPS showed little difference for Gaussian fit ensembles, no matter the ensemble size. Interestingly, CRPS for KDE fit ensembles was nearest the 'perfect' score when the ensemble size was 20; however, it continued to show decreases in CRPS (more skillful) as the ensemble size continued to decrease. In any case, CRPS required fewer forecast-sets to establish significant differences between the two dressing methods as compared to IS as the ensemble size shrank. It seems ensembles need to contain a minimum of approximately 15 members in order for IS to consistently achieve significance before CRPS. However, the ability of a KDE to resolve the underlying distribution for a given amount of ensemble members will depend partly on the distribution itself; thus these results and the effectiveness of each combination of scoring and dressing methods will also vary depending on the use case.

To summarize from an application standpoint: Table 1 implies that non-Gaussian dressing methods may not express any improvement in skill relative to Gaussian methods if CRPS is used as the scoring metric. The IS is much more sensitive to the structure of the PDF and can better reveal improvements from the use of non-Gaussian dressing methods. However, as illustrated by Fig. 9(b), the use of IS with the non-Gaussian dressing method considered here is much more sensitive to outliers.

**Table 1.** The mean score of postprocessed forecast distributions drawn with varying ensemble sizes. Rows indicate the scoring method and ensemble size, while columns indicate the forecast distribution used. The 'True' column refers to the 'perfect' score for each metric, where this is determined by evaluating on the true distribution (not the drawn ensembles). The 'No. required' column indicates how many 100-set forecasts are required before the sample mean Gaussian and KDE scores are distinct at the 95% confidence level for that particular skill score metric. Explicitly this is testing the variance of the mean of 100-set forecasts, where each set has the row's listed number of ensemble members.

| Scoring Method | True | Members | Gaussian | KDE | No. required |
|---|---|---|---|---|---|
| CRPS | 1.48 | 50 | 1.59 | 1.55 | 16601 |
| IS | 1.98 | 50 | 2.45 | 2.12 | 71 |
| | | | | | |
| CRPS | 1.48 | 20 | 1.59 | 1.51 | 2538 |
| IS | 1.98 | 20 | 2.46 | 2.20 | 144 |
| | | | | | |
| CRPS | 1.48 | 10 | 1.60 | 1.43 | 658 |
| IS | 1.98 | 10 | 2.48 | 2.37 | 1131 |

These results highlight the need to choose appropriate scoring and postprocessing methods to assess the skill in any bimodal forecast. New methods that are sensitive to distributional shape but do not penalize outliers so severely need to be developed before bimodality can be critically examined through a skill-based lens. Due to these difficulties, this study has not pursued further attempts to formally evaluate the effect on skill for the large occurrences of bimodality exhibited in the ECMWF dataset (Sect. 3). Instead, it focuses on identifying the presence of bimodality in the forecasts themselves, leaving for a future study the difficult question of validating this bimodality against observations.

## 6 Conclusions

The atmosphere is a highly chaotic system that is difficult to predict. While the assumption of Gaussianity leads to many simplifications, ensemble forecasts often exhibit non-Gaussian distributions, even in the 2-meter temperature field considered here. This work has systematically identified and characterized the presence of a specific case of non-Gaussianity, bimodality.

A statistical test for the presence of bimodality based on a KDE approach has been developed. This test identifies a 50-member forecast drawn from a Gaussian distribution as bimodal roughly 5% of the time, representing a general false-positive occurrence. Applying this test to five years of ECMWF ENS-extended forecasts, one finds this false-positive rate is exceeded over much of the globe during the entirety of the forecast, indicating that bimodality in these forecasts is a systematic and relatively common occurrence.

It has been found that bimodality is most frequent during weeks two and three of probabilistic forecasts as well as during the winter season, reaching values in excess of 30% in multiple regions including central South America, the North Atlantic

and the Antarctic Peninsula. In select regions such as around Indonesia and in the North Atlantic, bimodal forecasts occur in excess of 10% even during the first week of lead times.

The strikingly large occurrences over the Gulf Stream, the polar oceans and the equatorial oceans emphasizes the importance of the oceanic forcing while also suggesting very different processes associated with the development of bimodality in each region. Baroclinicity, sea ice boundaries and convection are possible candidates.

Although bimodal occurrence over land has been found to be less common than over the ocean, it is associated with modes that have greater separation in their temperature values. In fact, regions having an average mode separation of 10 - 15 $^o$C are not uncommon. Furthermore, some of these regions have modes that are 2 or 3 times as likely as compared to the distribution minimum, where the ensemble mean may lie. This has potentially drastic implications for bimodal forecasts fit with Gaussian distributions. Consider, for example, forecasts over southern Australia during its summer season. This region expressed bimodality in roughly 15-20% of its forecasts, for which the average mode separation is roughly 10 $^o$C and the relative probability ratio is around 3 or 4. If a wildfire forecast were to be issued, the non-Gaussian fit's warmer of the two modes may well surpass a critical temperature threshold representing high risk of wildfires. However, due to the large mode separation, a Gaussian distribution will likely have large variance. This will result in probabilities that are somewhat uniform and the risk of wildfires may be predicted to be very low.

It has been found that current ensemble scoring methods are not appropriate for assessing skill in non-Gaussian ensemble forecasts. CRPS is very weakly sensitive to the higher moments of distributions, and cannot clearly resolve the differences between Gaussian and non-Gaussian dressing methods even when the underlying process is strongly bimodal. The ignorance score proves to be a better alternative for non-Gaussian distributions, but is strongly sensitive to the tails of the distribution, and still requires a sufficiently large ensemble to identify a bimodal forecast as better than a Gaussian forecast when the underlying process is in fact bimodal.

The prevalence of bimodality in these ensemble forecasts suggests the need for improved methods to evaluate and post-process forecasts in the presence of bimodality. These are needed to properly assess and understand the role of bimodality and non-Gaussianity in the effective use of ensemble forecasts, especially on subseasonal timescales. Although non-Gaussian fitting methods prove to exhibit greater predictive power in some forecasts (Fig. 9), to do this for all forecasts may be computationally impractical. Thus the identification of when bimodality is most likely to occur is critical. Improved awareness of the nature of weather regimes, whose commonness may be supported by the high bimodality rates seen in this study, is a likely step forward to understanding bimodality in forecasts.

*Data availability.* The ECMWF ENS extended forecasts are accessible through the ECMWF website (https://www.ecmwf.int/en/forecasts). ERA-Interim data can be found at http://www.ecmwf.int/research/era.

*Code availability.* The code for data analyses and plots is based on the free Python software. Scripts are available upon request.

*Author contributions.* CB conducted analysis and prepared text. PH supervised, aided in the analysis and contributed to the writing of the text. AD and RP aided in the development of research questions and hypotheses. All authors were involved in the revision of the text.

*Competing interests.* The authors declare that they have no conflict of interest.

*Acknowledgements.* We thank Cornell University for funding and the use of materials and facilities, and the European Centre for Medium-Range Forecasts (ECMWF) for providing the dataset used. The authors also acknowledge the data center ESPRI/IPSL for their help in storing and accessing the data.

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
