# Peer review of "Bimodality in Ensemble Forecasts of 2-Meter Temperature: Identification"

_Weather and Climate Dynamics, 2021_

## Referee Comment (RC1)

**Review- Bimodality in Ensemble Forecasts of 2-Meter Temperature- Identification**

I have reviewed the paper "Bimodality in Ensemble Forecasts of 2-Meter Temperature- Identification" by Cameron Drew Bertossa, Peter Hitchcock, Arthur DeGaetano, and Riwal Plougonven.

In this manuscript the authors examine the non-Gaussianity (and specifically, bimodality) that arise in ensemble forecasts of T2m temperature. They show that fitting a Gaussian distribution to the ensemble PDF, which is commonly used, as well as the use of traditional skill scoring methods, performs poorly if bimodality is present. They then identify bimodality in ECMWF forecasts globally, and identify regions where bimodality appears strong.

This is a very interesting paper and I enjoyed reading it, it is well-written and presents interesting results. I think this is a very important issue that is often overlooked, and work of this sort has the potential to improve our forecast ability and understanding. I therefore recommend accepting the paper for publications in WCD and have only a few minor comments outlined below.

Comments:

- Line 35: What do you mean by "appropriate way"? not clear here. Do you mean the SVD analysis that you mention later?

- Eq.3- n is not defined here

- Lines 157 and 159: "skewed–> did you mean "shifted? that would be more appropriate here.

- Fig. 4b: I found this figure rather confusing. First, you only show the False Negative of the three distributions here, but from the text I was expecting to find also the False Positive. Also, why do you show the false positive of the Gaussian and not the false negative so we can compare it with the other distributions?

- Line 232: I'm not sure how you reached the 1.18 and 85% cutoff. Can you better explain this?

- Line 245: How do you determine the theoretical value of delta for the transition from unimodal to bimodal? From what I gather, the condition is $|\mu_2 - \mu_2| \leq 2\sigma$, regardless of $p$, which gives $\delta$=2 even if $p$ has now changed from $\frac{1}{2}$ to $\frac{3}{4}$. I assume you are not wrong, so I would love to understand what I am missing.

- Line 318 and Fig.3a: The fact that the mode ratio relies on the minimum (in the denominator), which is often close to zero, is a disadvantage as it can give high values just because it is low. Why not just take the ratio between to the peaks?

- General:

- What happens if there are actually more than two peaks?

-Can we somehow asses how much of the bimodality is originating from bimodality in the PDF of each forecast (e.g., perhaps in all the simulations the PDFs exhibit two maxima) as opposed to bimodality in the ensemble which results from different "regimes"? This is interesting because the spatial maps of delta look very much like regions of high temperature skewness.

---

## Author Response (AR1)

**The following formatting is used:**

RC comment is listed in blue

Response is listed in normal black text. In some cases figures are referenced in the response, these can be found at the end of the response document.

*Brief description and location (if not listed in comment) of changes in manuscript are listed in black italics.*

RC1:

- Line 35: What do you mean by "appropriate way"? not clear here. Do you mean the SVD analysis that you mention later?

"Appropriate way" does refer to the SVD analysis.

*No changes made in text*

- Eq.3- n is not defined here

In this case n refers to the number of members in an ensemble.

*n is now defined in text below equation 3.*

- Lines 157 and 159: "skewed–> did you mean "shifted? that would be more appropriate here.

Shifted is the correct term and has been changed in the text.

*Skewed is changed to shifted*

- Fig. 4b: I found this figure rather confusing. First, you only show the False Negative of the three distributions here, but from the text I was expecting to find also the False Positive. Also, why do you show the false positive of the Gaussian and not the false negative so we can compare it with the other distributions?

Since the distributions in blue are true bimodal distributions (refer to Fig. 4a), there would be no false-positive identification of bimodality for randomly sampled ensembles. The opposite is the case for the Gaussian distribution – there is no false-negative identification of bimodality for a distribution that is truly unimodal.

*No changes made in text*

- Line 232: I'm not sure how you reached the 1.18 and 85% cutoff. Can you better explain this?

1.18 and 85% refers to the same statistical requirement, where a mode probability ratio of 1.18 is equivalent to a minimum that is 85% (1/1.18) as likely as one of the peaks. This value was chosen because it appears to be a good compromise between reducing the false-positive identification of bimodality (around the inflection point of the dotted line in Fig. 4b) as well as minimizing the missed detections of true bimodal distributions (before the large increases in the blue distributions in Fig. 4b, note the different axes scales).

*Similar phrasing as above is added to the revised manuscript in order to explain the choice of the threshold.*

- Line 245: How do you determine the theoretical value of delta for the transition from unimodal to bimodal? From what I gather, the condition is $|\mu_2 − \mu_2| \leq 2\sigma$, regardless of p, which gives $\delta=2$ even if p has now changed from $\frac{1}{2}$ to $\frac{3}{4}$. I assume you are not wrong, so I would love to understand what I am missing.

The determination of whether the true underlying distribution was bimodal or unimodal was determined numerically based on the number of zeros of the first derivative. Since this was the same method being used to determine the critical points of the fit KDE distribution, this seemed a fair choice. The delta value representative of the transition from unimodality to bimodality was the value at which the number of zeros switched from 1 (one maximum in the distribution) to 3 (two maxima and one minimum).

*No changes made in text*

- Line 318 and Fig.3a: The fact that the mode ratio relies on the minimum (in the denominator), which is often close to zero, is a disadvantage as it can give high values just because it is low. Why not just take the ratio between to the peaks?

The representation of the mode probability as it is currently defined offers the advantage of understanding the potential gain from switching from a Gaussian fit to a non-Gaussian fit, since the probability at the minimum will be near that of the mean of the distribution. The mode probability ratio shows the great difference in probability that can occur at this minimum versus one of the bimodal distribution's modes. Additionally, the use of the ratio between the peaks will likely give similar results as using a mode probability ratio requirement for *both* peaks (as opposed to *either,* as is currently used), which was found to be associated with a much higher false-negative rate.

*No changes made in text*

General:

- What happens if there are actually more than two peaks?

This was briefly explored at the start of this study. However, trimodal distributions were found to occur in less than 5% of the forecast lead times which could largely be a result of false-positive detection.

*No changes made in text*

-Can we somehow assess how much of the bimodality is originating from bimodality in the PDF of each forecast (e.g., perhaps in all the simulations the PDFs exhibit two maxima) as opposed to bimodality in the ensemble which results from different "regimes"? This is interesting because the spatial maps of delta look very much like regions of high temperature skewness

This is an interesting point that is currently being examined in a follow up study. It is unsure if this has a definite answer right now. However, if the underlying distribution itself were to be bimodal, it is one's intuition to believe that these distributions would then remain bimodal as they approach the climatological bimodal PDF. This has been found to *not* be the case in many forecasts examined (though an exact quantification is not known), one example being the forecast from Fig. 1d. You may refer to figure 2 below and the corresponding discussion surrounding RC2's major point #3 for more detailed insight into this. Additionally, we may refer to (Tamarin-Brodsky et al. 2019) Figure 1c to see that there are regions during boreal winter in which the minima in bimodality do not align with the minima in skewness, that is not to say they skewness and bimodality are unrelated but rather there is some amount of uniqueness. Specific examples of uniqueness include west of South America (a minima in bimodality but large skew) east of Greenland (maxima in bimodality but little skew) and west of Australia (minima in bimodality but large skew).

*Similar discussion regarding uniqueness between skew and bimodality is added to lines 278-283 in the text.*

RC2:

1. The test is built in a very simple framework of synthetic data generated by a mixture of Gaussians. Data from the ECMWF are not purely random variables and one may wonder whether this type of approach is valid in a more realistic context. I am surprised that the test has not been validated on more sophisticated systems like the 3-variable Lorenz model (1963) displaying bimodality for certain variables, and also on spatially distributed systems like the Lorenz 95 system (1995). This would allow to evaluate the ability of the test to indeed detect bimodality, and to test the sampling issue in detecting false-positives. This should be done on these models or other similar types of models.

As suggested, a timeseries generated by the Lorenz-63 model is created and the criteria currently being used (based on mode probability ratio and membership) is tested against multiple underlying distributions to compare the false-positive and false-negative rate. With this model we generated a very large ensemble (10,000 members) and allowed it to evolve to serve as the 'true distribution'. Then at discrete timesteps we can repeatedly draw 50-member ensembles and apply our criteria to determine the false-positive/negative occurrences for various distributions. This is a similar process as to what was done for the analysis surrounding Figure 4. Below is a figure representing the Lorenz-63 timeseries used and the distributions drawn (first figure below). One bimodal distribution is used (testing false-negatives) as well as two unimodal distributions. One of the unimodal distributions is drawn early into the ensemble spread (representative of early into a forecast), has a small standard deviation and appears to be nearly Gaussian. The false-positive percentage is roughly 3%. The other unimodal distribution is drawn much later into the 'forecast', has a much larger spread and looks less Gaussian. However, the false-positive percentage is still only 10%. These results agree well with the current synthetic study. Additionally, we have marked where the distribution is bimodal (red ticks) and where the KS statistic deems it the same as the climatological distribution (blue ticks), which was determined by integrating for many time steps. While the synthetic data was generated in a simplistic way, even the two models proposed have far fewer degrees of freedom than that of the atmosphere. That is to say, it will be difficult to exactly simulate a 'true' distribution of the atmosphere no matter the method used to generate the synthetic data. In any case, this figure and discussion surrounding it is added at the end of the current manuscript's synthetic study in order to increase the reader's confidence in the results.

*Figure inserted after Figure 4 which depicts a Lorenz system. Discussion can be found in lines ~260-275. This is used as a means of increasing the reader's confidence in the synthetic results, however, is also used as a simple framework for how bimodality is expected to develop.*

2. Bimodality is detected in very active zones of the atmosphere such as for instance in panel (d) of Figure 1. When I look at this figure, it seems to me that after 5 days there is no skill anymore, and after that period there is a random succession of unimodal, bimodal (and multi-modal) situations. At first sight I would say that this is a problem of sampling, and one can have on such a large range of possible values (from -15 to 5 °C) random clustering of ensemble members. In order to check whether it is not an artifact of sampling, one can pool several successive ensembles, for instance the ensembles started 12 hours before and after the nominal ensemble (or another grouping of ensembles). In that way one can get a set of 153 ensemble members and see if the bimodality detected is still present.

This process was done for the forecast in Figure 1d. We isolated analysis to lead times that correspond to days 5-20 days from the forecast in 1d, this is where bimodality seems to be present in the current example. Only forecasts whose lead times would correspond to weeks 2&3 were used (days 7-21) as early and very late into the forecast we'd expect ensembles to be nearly unimodal due to little or great spread, respectively. For this example, this resulted in using the three forecasts initialized prior and one forecast initialized after the June 16[th] forecast. This results in about 250-member ensembles at a given lead time. A contour plot for days 5-20 corresponding to the June 16[th] forecast is attached below (second figure below). While we derived the bimodal criteria for a 50-member ensemble, we linearly scaled the membership requirement to still demarcate where the large ensemble is deemed bimodal; while this hasn't been critically tested it still offers a useful comparison. Where a non-Gaussian fit has higher skill than Gaussian is also demarcated. Both of these (and those points in which there is overlap) align with many of the points previously marked in the 50-member ensemble. Additionally, we see two modes still clearly present in the contours at many lead times. It appears that the colder mode is actually biased too warm relative to the observations. More generally, it should be noted that many of those locations who are deemed 'bimodal' align with high skill locations (red markers in Figure 1d.). Additionally, observations seemingly flip between modes in the figure, with very little time in between. Note, for example, just after a lead of 15 days when the observation flips to a much colder temperature just after a second mode reemerges in the fit KDE (this is even more strongly supported in the 250-member ensemble). The combination of these two above points would be surprising for a sampling error. While the 250-member ensemble was a helpful check for this particular forecast, this process was not conducted for the rest of the forecasts that were found to be bimodal. This has led us not to include this additional figure or analysis in the revised manuscript.

*No changes made in text.*

3. The authors mention in the discussion (lines 325-330) the convergence toward the asymptotic distribution, and indicate that the bimodality is more a transient process. This should be checked and compared with the asymptotic climatological distribution. We should expect the convergence toward the climatology.

This point was tested using hindcasts for the same forecast location and day-of-year initialization date shown in Figure 1d, with one hindcast per year for 20 years. These hindcasts are smaller ensembles with 10 member each, however, have the same temporal resolution (6 hourly for 46 days) as the forecast. A 'climatology' was built using the observations (refer below to minor point #1 for how these are derived) from each of the hindcast lead times. Since we were only concerned with seeing if the end of forecast lead times were approaching climatology, we picked weeks 4 and beyond to focus on in the hindcasts (and thus then also the observation climatology). We then applied a 2-sided Kolomogorov-Smirnov test to see if these distributions were significantly different. Refer to the third figure below. The p-value was found

to be essentially 0 meaning we reject the null hypothesis that the distributions are identical. This means that forecasts are not the same as climatology even at the very large forecast lead times. This does, however, mean that the ensemble forecast is not simply a sample of the climatological distribution and the bimodality being expressed is particular to the forecast (rather than the climatology itself) even at long lead times. As to why the distributions are still significantly from climatology at 4+ weeks: this may be explained in part by interannual variability such as ENSO or MJO which could cause forecasts to diverge from climatology even at very long lead times. In the revised manuscript we have been more careful in declaring that the long lead time distributions are the same as climatology. The use of the phrase 'approaching climatology' may be more accurate.

*In reference to this process the authors are careful to use 'approaches climatology'.*

4. In the discussion section (5), the authors discuss the possible origin of the bimodality, in particular related with the interaction with the ocean. Would the authors mean that the bimodality is related to a low-frequency variability which has skill at longer lead times? It would certainly be useful to investigate in greater details this in a specific region like the one presented in Figure 1d and see how is the convergence toward climatology through the use of skill scores, and see what is the difference between the events that are truly bimodal and the other ones. And also relate these to some specific processes (the presence of sea ice for instance).

Questions similar to this are being investigated in a follow up manuscript. Bimodality in part appears to be related to boundary forcings by the ocean, both in terms of SST gradients and sea ice dynamics. However, this analysis is beyond the objectives of this manuscript (identifying where it is present rather than fully explaining it).

*No changes made in text*

5. In the first part of the manuscript the authors address the problem of scoring and statistical postprocessing when the distributions are bimodal. In particular, the ability of scoring rules to detect bimodality. But at the end of section 2, the authors indicate the difficulty to detect bimodality based on the scoring rules, and they then move to the analysis of bimodality through the development of a simple statistical test. This section 2 looks unrelated with the rest of the manuscript and I think that this part can be considerable shortened and either placed in an Appendix, or as a few paragraphs in the introduction.

We believe these results give important insights for the forecast scoring community. Rather than moving this to an appendix, text has been added in the abstract, introduction and conclusions to incorporate the skill scoring section better into the manuscript.

*A single line added at the end of the abstract emphasizing the need for improved skill scores. A more activate voice is used in lines 191-195 to emphasize the skill score aspect of this study.*

Minor points:

1. In Figure 1, panel a, the authors show the observation in yellow. On what is this observation based on? And there is of course an observational error which is not taken into account in these plots and in the comments. The authors indicate under-dispersion of the ensemble, but is it really under-dispersed? This cannot be said from a single ensemble forecast.

While we agree that a single ensemble forecast does not prove under dispersion, it was cited in the previous paragraph that ensemble forecasts are historically under-dispersed. This specific forecast may show an example of that under-dispersion, specifically how the observation lays far outside the forecast distribution for a majority of the early lead times. The validating 'observations' in Fig. 1 are drawn from the ERA Interim reanalysis (see Dee et al. 2011 QJRMS DOI:10.1002/qj.828). We haven't considered the error in the reanalysis explicitly, but it is unlikely to affect the discussion, which has more to do with the behavior of the ensemble forecasts.

*Information and reference to Dee et al (2011) is added to the revised manuscript (line 50).*

2. Lines 150-155. The authors mention the use of statistical postprocessing. It is not clear at all how it is done. This should be explained.

This has been expanded upon in the revised text. Briefly, this is an optimization for a bias parameter and a dispersion parameter which minimizes the fit distribution's skill score.

*This is now explained explicitly in the text at line 155. A shift to the mean and dispersion is applied for optimization.*

3. Line 220. Please add that material in a supplementary. Also the material discussed at lines 170-175 if section 2.3 is kept somewhere (either in an Appendix or in a supplement).

We feel that this additional material adds confusion for the reader without adding to the text. Namely, the advantages of the membership requirement (line 220) is expressed already in Figure 4. Similarly with the material presented in 170-175, the additional material is not unique to what is already presented in the table; the authors feel that stating this fact is sufficient.

*No changes made in text*

[Figure]

[Figure]

(d)

June 16, 2016

(-65.7° N, 232.2° E)

both

$P_{KDE} > P_{Gauss}$

bimodal

---

## Editor Decision (ED1)

wcd-2021-33
Editor decision – comments to the authors

**Bimodality in Ensemble Forecasts of 2-Meter Temperature: Identification**

by C. Bertossa et al.

Dear Cameron Bertossa and colleagues

Many thanks for your latest revisions and for addressing the points raised by the reviewers in more detail. I am happy to accept your paper for publication in WCD subject to technical corrections, as suggested below. Congratulations and thank you for submitting your paper to WCD! Your paper very nicely illustrates and discusses the importance of bimodality in extended-range ensemble predictions.

Line 44: "… are shown in Fig. 1." To me it would be helpful to add something like "at selected grid points". Also maybe mention in the figure caption or in the text whether you look at 6-hourly T2m values or daily means.

Line 50: I am not a native speaker and therefore some of my language suggestions might be inappropriate. I just mention a few sentences where I found the wording not ideal. A first example is on line 50. Here I would prefer "… whereas in another forecast for/at the same location …" and in line 52 "… which show the same forecast but at different locations."

Caption Figure 1 last line "times which" → "times when"

Line 74: describe → described

Eq. 2: in latex use \log instead of log

I would like to come back to one of the original comments from reviewer 2 about Sect. 2. The reviewer mentioned that this section, although very interesting, is not directly related to main story of the paper. Now reading the entire revised paper I had a similar impression. After the introduction I was curious to see where bimodality occurs (i.e., I was curious about the results shown in Sect. 4) and I could not fully understand why I need to go through a relatively long and complicated section about ensemble scoring. Or in other words, maybe it would be easier for me as a reader to absorb this material after having seen where bimodality occurs (and how often, etc.). I ask you to reconsider whether Sect. 2 should remain where it is or whether shifting it to an Appendix would be more appropriate for the WCD readership. I leave this decision up to you and just wanted to share with you my impressions about Sect. 2 when reading the paper from A to Z.

Line 192: either "itself; thus" or "itself. Thus"

Line 210: Fig. 3 → Fig. 3a

Line 220: you explain the notation N (Gaussian distribution) here, but you already used it in Fig. 2.

Section 3 is long and contains different aspects. I suggest introducing subsections, with new subsections starting at lines 205, 264 and 290.

Line 267: "relative to one another"

Caption Fig. 5: I think this should read "… timesteps when the ensemble has two modes"

Line 307: dates should be in format 03 December 2015 etc.

Caption Fig. 6: why "versus"? I suggest "… times, (b) in weeks 2-3, and (c) in weeks 4-5 …"

Caption Fig. 7: strange wording, I suggest "(a) is for lead times in week 1, (b) in weeks 2-3, and (c) in weeks 4-5 …"

Lines 340 and 343: no need to start new paragraph

Figure 8: These panels are identical to the ones shown in Figs. 6b and 7b! I am not sure if maybe the intention was to show the entire year in Figs. 6 and 7? If not, then there is no need to include Fig. 8.

Line 357: not sure, is "as pronounced of a seasonal dependence" proper English?

Line 358: should read "… this may be affected to some extent (with t) …"

Line 365: ; should read .

Line 395: should read "8 and 9". Then: "Tamarin-Brodsky et al. (2019) depicted …"

Line 400: should read "… found in Tamarin-Brodsky et al. (2019, their Fig. 1c) …"

Line 420: not sure, are the formulations "a property opposite most the globe" and "ITCZ position based off rainfall climatology" proper English? (At least I don't understand these sentences.)

Line 435: "can not" → "cannot"

References: Copernicus journals use journal abbreviations and include a DOI. Please check in published WCD papers and adapt your list of references accordingly.

I am looking forward to receiving the final version of your manuscript.
With best regards,
Heini Wernli

---

## Author Response (AR2)

**CoEditor Comments:**

Many thanks the revisions of your paper. I sent the new version to the 2nd reviewer who suggested major revisions. As you can see in the response, the reviewer appreciates your responses to the issues about the sampling and the convergence toward the climatology, but strongly suggests to include them (maybe in condensed form) in the paper. I also had the impression that given the fact that the reviewer suggested "major revisions" you actually changed little in the paper. I suggest that you follow the reviewer's advice and include more insights about the two mentioned aspects.

I am looking forward to the updated version of your paper.

**Authors' Response:**

Thank you for the feedback and kind words. We have revised the manuscript, including discussion surrounding the distinction from climatological behavior. This can be found on line 290 and beyond. We choose, however, to supplement some discussion in our author's response in order to emphasize the difficulty of answering this question beyond what is now provided in the text:

"If the occurrence of bimodality is a result of the climatological distribution itself, one would expect occurrence rates to asymptote at very long forecast lead times as the forecasts approach the climatological distribution. However, if bimodality is occurring due to particular atmospheric states, one would expect there to be some 'peak' in bimodal frequency at some function of lead time prior to an asymptotic behavior, this 'peak' period would be representative of the second period in Figure 5.

This in itself, however, is a difficult question to answer systematically and requires a study of its own. The reason being, there may be some skill to forecasts even at very long lead times (weeks 4+) which cause them to not yet approach the climatological distribution (and thus asymptotic behavior in bimodal frequency is not yet exhibited). This would be expected to be especially prevalent in a location like the tropics, where the effect of ENSO has a signal that spans beyond the sub-seasonal timescale. Thus, ensemble forecasts that are both able to simulate particular weather events (which may cause the development of bimodality), as well as span long enough (perhaps much longer than the 6+ weeks of ECMWF), would be needed globally. This on top of the fact that large enough ensembles are needed to exhibit bimodality in the first place (refer back to Sect.2) and that a great number of forecasts are desired to achieve a sufficient sample size for the study, a constraint by computational power is created."

Furthermore, as a note, we have revised some content within the skill scoring section 2.3. Mainly in regards to being more explicit in our discussion and the expansion of Table 1 with ensembles of different sizes. We hope this offers a clearer discussion of a rather important point to the skill scoring community.